# Identifying key signs of motor neurone disease in primary care: a nested case–control study using the QResearch database

Xue W Mei ,[1] Judith Burchardt,[1] Tom A Ranger ,[1]
Christopher J McDermott ,[2] Aleksandar Radunovic,[3] Carol Coupland,[1,4]
Julia Hippisley-Cox [1]

¹Nuffield Department of Primary Care Health Sciences, University of Oxford, Oxford, UK
²Sheffield Institute for Translational Neuroscience, University of Sheffield, Sheffield, UK
³Barts MND centre, Barts Health NHS Trust, London, UK
⁴School of Medicine, University of Nottingham, Nottingham, UK

**Correspondence to**
Professor Julia Hippisley-Cox;
julia.hippisley-cox@phc.ox.ac.uk

## ABSTRACT

**Objective** To confirm the symptoms and signs for motor neuron disease (MND) in the Red Flag tool; to quantify the extent to which the key symptoms and signs are associated with MND; and to identify additional factors which may be helpful within the primary care setting in recognition of possible MND and triggering timely referral to neurology specialists.

**Design** A nested case–control study.

**Setting** 1292 UK general practices contributing to the QResearch primary care database, linked to hospital and mortality data.

**Participants** Baseline cohort included 16.8 million individuals aged 18 years and over without a diagnosis of MND at study entry and with more than 3 years of digitalised information available. The nested case–control data set comprised of 6437 cases of MND diagnosed between January 1998 and December 2019, matched by year of birth, gender, general practice and calendar year to 62 003 controls.

**Main outcome measures** Clinically recognised symptoms and signs of MND prior to diagnosis and symptoms and factors which are relevant in primary care setting.

**Results** This study identified 17 signs and symptoms that were independently associated with MND diagnosis in a multivariable analysis. Of these, seven were new to the Red Flag tool: ataxia, dysphasia, weight loss, wheeze, hoarseness of voice, urinary incontinence and constipation. Among those from the Red Flag tool, dysarthria had the strongest association with subsequent MND (adjusted OR (aOR): 43.2 (95% CI 36.0 to 52.0)) followed by muscle fasciculations (aOR: 40.2 (95% CI 25.6 to 63.1)) and muscle wasting (aOR: 31.0 (95% CI 19.5 to 49.4)). Additionally, the associations between MND diagnosis and family history, dropped foot, focal weakness and sialorrhoea remained robust after controlling for confounders. Patients who reported symptoms indicative of damage to the lower brainstem and its connections were diagnosed sooner than those who presented with respiratory or cognitive signs.

**Conclusion** This is the first study that has identified, confirmed and quantified the association of key symptoms and signs with MND diagnosis. In addition to known factors, the study has identified the following new factors to be independently associated with MND prior to diagnosis: ataxia, dysphasia, wheeze and hoarseness of voice. These findings may be used to improve risk stratification and earlier detection of MND in primary care.

## STRENGTHS AND LIMITATIONS OF THIS STUDY

⇒ This is the largest nested case–control study derived from a representative population-based cohort looking at signs and symptoms related to motor neuron disease (MND).

⇒ We were able to collate an extensive list of signs and symptoms and to examine the association of each to MND diagnosis.

⇒ We were not able to determine the time point, number and types of specialist referrals.

⇒ We do not know about completeness and accuracy of the recording, since not all patients with symptoms will seek medical attention and not all symptoms will be recorded at the general practitioner (GP) consultations.

⇒ Some observed associations may be due to unmeasured confounding or variations in coding practices at the GPs.

## INTRODUCTION

Motor neuron disease (MND) causes progressive neuromuscular weakness that may first present as isolated and unexplained symptoms. MND causes the deaths of 1 in 350 men and 1 in 470 women in the UK,[1] although there is emerging evidence that the incidence might be higher than previously thought.[2] In Europe, the annual crude incidence rate was estimated to be 2.7 per 100 000 person-years.[3] MND is difficult to recognise in primary care since it is both a relatively uncommon disease and the clinical presentations of early symptoms are sporadic and non-specific. As a result, patients may delay consulting their general practitioners (GPs) and the GPs may not attribute the symptoms to MND.

Previous work has demonstrated that both patient factors (eg, delayed presentation to GP) and healthcare practice factors (eg, referrals to non-neurology specialists) contribute to delays in MND diagnosis, hence patients' access to disease management.[4] The length of diagnostic delay has also been found to be mostly between 10 and 16 months from symptom onset to diagnosis.[4]

In 2014, the MND Association and the Royal College of General Practitioners collaborated to produce a Red Flag tool for MND,[5] which is designed to improve timely referrals to neurology specialists from primary care consultants. This tool was developed based on interviews with MND specialists and details a number of symptoms and signs, which could be indicative of MND but has not been verified by analysis of clinical data. Therefore, the aims of our study are to confirm the symptoms and signs in the Red Flag tool; to quantify the extent to which the key symptoms and signs are associated with MND as well as to identify additional factors which may be helpful within the primary care setting in recognition of possible MND and triggering timely referral to neurology specialists.

## METHOD
### Study design
We conducted a nested case–control study in a large UK primary care cohort using the QResearch database (V.44). QResearch is an anonymised, large open cohort database, which prospectively collects routine health data from general practices in the UK that use the Egton Medical Information Systems computer system (used by 55% of GPs in the UK). The data include individual-level data for demographics, lifestyle factors, medical diagnoses, prescriptions, specialist referrals, medical examinations and examination results and are linked to Hospital Episode Statistics (HES) records and Office of National Statistics (ONS) death registry records.

### Ascertainment of cases and controls
The base cohort included individuals aged 18 years and over registered during the study period (1 January 1998 to 31 December 2019) without a diagnosis of MND at study entry. The study entry date was defined as the latest of any of the following: 18th birthday; date of registration with the practice plus 1 year; date on which the practice computer system was installed plus 1 year or the beginning of the study period (1 January 1998). The cohort was followed up until the earliest of any of the following: the date of MND diagnosis; date of 100th birthday; date of death; date of leaving the practice or the study end date (31 December 2019).

Cases were defined as patients in the study cohort with a new diagnosis of MND on any of the GP record, HES record or ONS death registry during follow-up. The following Read codes were used to identify the cases from the GP records: F152 (MND), F1520 (amyotrophic lateral sclerosis (ALS)), F1521 (progressive muscular atrophy), F1522 (progressive bulbar palsy), F1523 (pseudobulbar palsy), F1524 (primary lateral sclerosis) and F152z (MND not otherwise specified). We also used ICD10 code G122 (MND) to identify further records from HES and ONS death registry.

Controls were matched to cases by sex, practice, year of birth and calendar year using the incidence density sampling method.[6] Each control was allocated an index date which was the date of first MND diagnosis of their matched cases. Each case was matched with up to 10 controls who were alive, registered and without a diagnosis of MND at the index date. Cases and controls were excluded if they had less than 3 years of computerised data available at the index date, to ensure that the data are complete for a minimum of 3 years prior to index date.

## SYMPTOMS
Red flag symptoms included clinically recognised symptoms and signs of MND.[5 7] The list of existing factors is as following: bulbar features—dysarthria, dysphagia, sialorrhoea or excessive salivation and tongue fasciculations; limb or muscle features—focal weakness, falls, foot drop, muscle wasting, muscle fasciculations or twitching, cramps, sensory impairment and muscle stiffness; respiratory features—orthopnoea, dyspnoea, shortness of breath on exertion, sleep problems, tiredness or fatigue and early morning headache; cognitive features—behavioural change, emotional lability, depression, hallucinations and confusion as well as appetite loss and family history of MND as supporting factors. Additionally, we included potential symptoms such as weight loss, ataxia, dry mouth, heaviness of legs and difficulty climbing stairs, which were identified through literature searches[8–10] and a search of Read/SNOMED-CT codes for symptoms recorded prior to diagnosis in a pilot study involving a sample of patients diagnosed with MND in JB's general practice. These were included because although not directly established signs/symptoms of MND, these may provide valuable information in identifying additional patients who may benefit from neurologist referral in the primary care setting. We also included the following list of conditions to serve as negative controls since these are not usually considered to be associated with MND: diplopia, ptosis, bladder and bowel problems such as incontinence and constipation as well as sexual dysfunction. All symptoms and signs were identified through searches of the GP records via corresponding Read/SNOMED-CT codes. For the primary analyses, only signs and symptoms that were last recorded between 3 months to 5 years prior to the index date were included, as symptoms occurring more than 5 years ago were considered unlikely to be relevant to MND and those recorded less than 3 months prior are likely to be reflective of the average waiting period between specialist referral and formal diagnosis. We also excluded very rare signs and symptoms (defined as <5 ever recorded counts in either cases or controls). Table 1 has a summary of the factors included in the analysis and the source of the information.

**Table 1** Factors included for analysis and the source from which they were identified*

| Symptoms and signs | Source/reason for inclusion | Confirmed on analysis† |
|---|---|---|
| Bulbar features | | |
| Dysarthria | MNDA | UV, MV |
| Dysphagia | MNDA | UV, MV |
| Sialorrhoea/excessive salivation | MNDA | UV, MV |
| Tongue fasciculations | MNDA | UV |
| Limb features/muscle weakness | | |
| Focal weakness | MNDA | UV, MV |
| Falls | MNDA | UV, MV |
| Foot drop | MNDA | UV, MV |
| Muscle wasting | MNDA | UV, MV |
| Muscle fasciculations or twitching | MNDA | UV, MV |
| Cramps | MNDA | UV, MV |
| Sensory impairment | MNDA | No |
| Muscle stiffness | NICE | UV, MV |
| Respiratory features | | |
| Orthopnoea | MNDA | No |
| Dyspnoea | MNDA | UV |
| Shortness of breath on exertion | MNDA | UV |
| Sleep problems | MNDA | UV |
| Tiredness/fatigue | MNDA | UV |
| Early morning headache | MNDA | No |
| Cognitive features | | |
| Behavioural change | MNDA | No |
| Emotional lability | MNDA | UV |
| Depression | MNDA | UV |
| Hallucinations | NICE | UV |
| Confusion | NICE | UV |
| Supporting factors | | |
| Family history of MND | MNDA | UV, MV |
| Appetite loss | NICE | UV |
| Additional factors from literature review/pilot study | | |
| Alcoholism | Literature | No |
| Weight-loss | Literature | UV, MV |
| Ataxia | Pilot study | UV, MV |
| Wheeze | Pilot study | UV, MV |
| Difficulty climbing stairs | Pilot study | UV |
| Dry mouth | Pilot study | UV |
| Dysphasia | Pilot study | UV, MV |
| Hoarseness of voice | Pilot study | UV, MV |
| Factors not considered supportive of diagnosis by MNDA | | |

Continued

**Table 1** Continued

| Symptoms and signs | Source/reason for inclusion | Confirmed on analysis† |
|---|---|---|
| Urinary frequency increase | Negative control | UV |
| Urinary incontinence | Negative control | UV, MV |
| Urinary retention | Negative control | UV |
| Nocturia | Negative control | UV |
| Dysuria | Negative control | No |
| Constipation | Negative control | UV, MV |
| Faecal incontinence | Negative control | No |
| Impotence | Negative control | No |
| Diplopia | Negative control | No |
| Ptosis | Negative control | No |

*The study was matched on age and gender therefore they were not included in the analysis model.
†Indicates the analysis model used to find the associations. All models were additionally adjusted for body mass index, Townsend scores in quintiles, smoking status, alcohol consumption behaviour and self-reported ethnicity.
MV, multivariate; UV, univariate.

### Statistical analysis

We estimated the strength of association of the red flag signs and symptoms with MND diagnosis, expressed as ORs, using conditional logistic regression. We performed both univariate and multivariable analysis, adjusted for the following confounders: the most recently recorded body mass index (BMI) prior to index date, latest records of smoking status and alcohol consumption, Townsend deprivation score in quintiles and self-assigned ethnic groups. We assumed missing data on BMI, smoking, alcohol consumption, Townsend scores and ethnic groups were missing at random and imputed them using multiple imputation by chained equations.[11–13] We created 20 imputed data sets and the imputation model included demographic variables, comorbidities or disease history (ie, ever diagnosis of one or more of the following: cardiovascular diseases, stroke, diabetes (type 1 and 2), dementia, Parkinson's disease, myasthenia gravis and multiple sclerosis), all symptoms and signs that were included in the final regression model as well as the case–control indicator. The ORs from each imputed dataset were combined to form the final adjusted ORs using Rubin's rule.[14] Only signs and symptoms that reached statistical significance (two-tailed p<0.05) in the univariate analysis were included in the multivariable analysis. All statistical analyses were performed using Stata V.16 (Stata Corp, College Station, Texas).

### Power calculations

A feasibility study identified over 5000 patients with MND in the last 20 years in the QResearch. With 5000 cases and 10 matched controls per case, we will be able to detect an OR of 1.26 or more for a symptom recorded in 5%

of controls, with 90% power and 1% significance. For a symptom recorded in 1% of controls, we will be able to detect an OR of 1.59 or more.[15]

## Patient and public involvement

This study was initiated by the MND Association, which is a charity representing patients with MND. Apart from this, patients and public were not involved in design and development of the study or interpretation of the study outcome but will be consulted regarding the dissemination of the results.

## RESULTS

### Descriptive characteristics of the study cohort

The base cohort comprised of 16 799 992 participants aged between 18 and 100 years. During a total of 112 003 453 person-years of follow-up, we found 6437 incident cases of MND that were eligible for inclusion, using records from general practice, HES and death registry databases. We were able to identify a total of 62 003 birth year, gender, practice and calendar year-matched controls. 56.8% of the MND cases were men and mean (SD) age at diagnosis was 69.6 (12.2) years.

Table 2 is a summary of the demographic characteristics of the cases and controls. The ethnicity recording was more complete in controls (71.3%) than in cases (55.4%); however, completeness of Townsend deprivation score, alcohol, smoking and BMI were similar between the two groups. There was no significant difference between cases and controls in terms of alcohol consumption or smoking status. However, compared with the controls, the cases had lower BMI (p<0.001) and were more likely to be living in less deprived areas based on Townsend score quintiles (p=0.013).

### Association of red flag signs and symptoms to MND diagnosis

The types of symptoms and signs as well as their adjusted estimates are presented in figure 1 (univariate) and figure 2 (multivariate). Table 1 also contains the list of factors examined, source from which they were obtained, and whether they were statistically significant in the analysis. Thirty-three signs and symptoms reached our prespecified level of statistical significance (two-tailed p<0.05) in the unadjusted univariate analysis, and these remained significant after adjustment for BMI, alcohol consumption, smoking, deprivation and self-reported ethnicity (figure 1). These were collectively included in a multivariable model for estimation of their independent associations with MND diagnosis. Dysarthria was the strongest independent predictor of MND diagnosis (OR 43.2, 95% CI 36.0 to 52.0) followed by muscle fasciculations (OR (95% CI) 40.2 (25.6 to 63.1)), muscle wasting (OR (95% CI) 31.0 (19.5 to 49.4)), and dropped foot (OR (95% CI) 14.8 (11.3 to 19.3)). Some respiratory symptoms such as shortness of breath and dyspnoea, as well as cognitive symptoms such as depression and confusion, though statistically significant in univariate analysis, were no longer associated with MND after controlling for other symptoms and signs. We also found the following symptoms, which were not

included in the Red Flag tool but were found in GP records, to be independently associated with MND diagnosis: hoarseness of voice (OR (95% CI) 3.26 (2.82 to 3.77)), dysphasia (OR (95% CI) 2.85 (1.92 to 4.23)), ataxia (OR (95% CI) 4.82 (3.58 to 6.48)), wheeze (OR (95% CI) 1.15 (1.01 to 1.32)) and weight loss (OR (95% CI) 2.20 (1.92 to 2.52)).

We also observed independent associations with MND for two of the symptoms not generally considered to be associated with MND: constipation (OR (95% CI) 1.45 (1.31 to 1.60)) and urinary incontinence (OR (95% CI) 1.28 (1.12 to 1.47)).

Overall, symptoms from bulbar dysfunction (ie, dysarthria, dysphagia, sialorrhoea and tongue fasciculations) and those appearing in the limbs are most strongly suggestive of subsequent MND diagnosis. Symptoms and signs occurring in the respiratory and cognitive systems were not statistically significant when evaluated in combination with others.

### Time from earliest symptom presentation to MND diagnosis

Duration from the earliest symptom presentation to MND diagnosis varied considerably by feature and location of the signs and symptoms. Patients presenting with symptoms and signs associated with damage of bulbar region and limb functions received their diagnoses sooner than those presenting with respiratory or cognitive symptoms and signs (eg, median days from symptom presentation to diagnosis: dysarthria 145.5 days (IQR 68.5–296 days) versus orthopnoea 299 days (IQR 42–496 days)). Among those clinical features, unusual and sporadic signs such as dysarthria and muscle fasciculations appeared to be the key indicators to speedy neurologist referral. Time from earliest clinical symptom presentation to MND diagnosis (median days, IQR) is presented in figure 3.

## DISCUSSION

### Principal findings

In this very large population-based nested case–control study covering a population of over 16 million adults, we were able to demonstrate that the MND Association recommended Red Flag signs and symptoms,[5] particularly those affecting bulbar and limb regions, are independently predictive of subsequent MND diagnosis. Among our study of 6437 MND patients and 62 003 matched controls, the symptom with the largest association with MND was dysarthria followed by muscle fasciculations and muscle wasting. Additionally from the GP records, we found the following symptoms to be independently associated with MND: hoarseness of voice, dysphasia, ataxia, wheeze and weight loss. We also found a higher proportion of MND cases with documented constipation and urinary incontinence than the controls.

Consistent with previous studies,[16 17] we found patients showing bulbar onset symptoms such as dysarthria and dysphagia were diagnosed on average 4–5 months sooner than those with limb onset symptoms. Interestingly, in our cohort, patients presenting with muscle fasciculations were diagnosed most quickly, which is likely as a result of timely and adequate specialist referral by their managing GPs. This is consistent with a qualitative study conducted

**Table 2** Demographic characteristics of case patients with motor neuron disease and age, sex practice, calendar time matched controls

| Characteristic | Study participants, number (column %) | |
| --- | --- | --- |
| | Controls (n=62003) | Cases (n=6437) |
| Women | 27058 (43.6) | 2781 (43.2) |
| Men | 34945 (56.4) | 3656 (56.8) |
| Age at diagnosis/index date (mean(SD)), y | 69.1 (11.9) | 69.6 (12.2) |
| Age at study entry | | |
| <50 | 10700 (17.3) | 1078 (16.7) |
| 50–54 | 6556 (10.6) | 651 (10.1) |
| 55–59 | 8016 (12.9) | 793 (12.3) |
| 60–64 | 9678 (15.6) | 976 (15.2) |
| 65–69 | 9734 (15.7) | 982 (15.3) |
| 70–74 | 8085(13) | 852 (13.2) |
| 75–79 | 5771 (9.3) | 627 (9.7) |
| 80 + | 3463 (5.6) | 478 (7.4) |
| Ethnicity recorded | 44211 (71.3) | 3563 (55.4) |
| Ethnicity | | |
| White | 41106 (66.3) | 3281(51) |
| Indian | 700 (1.1) | 69 (1.1) |
| Pakistani | 349 (0.6) | 32 (0.5) |
| Bangladeshi | 174 (0.3) | 19 (0.3) |
| Other Asian | 322 (0.5) | 30 (0.5) |
| Caribbean | 537 (0.9) | 48 (0.7) |
| Black African | 373 (0.6) | 28 (0.4) |
| Chinese | 132 (0.2) | 4 (0.1) |
| Other | 518 (0.8) | 52 (0.8) |
| Not recorded | 17792 (28.7) | 2874 (44.6) |
| Townsend deprivation score, quintiles | | |
| 1 (least deprived) | 20410 (32.9) | 2205 (34.3) |
| 2 | 16131(26) | 1613 (25.1) |
| 3 | 11533 (18.6) | 1174 (18.2) |
| 4 | 8263 (13.3) | 795 (12.4) |
| 5 (most deprived) | 5587(9) | 640 (9.9) |
| Townsend deprivation score not recorded | 79 (0.1) | 10 (0.2) |
| BMI not recorded before index date | 11352 (18.3) | 1124 (17.5) |
| BMI (mean(SD)) | 26.6 (4.6) | 26.2 (4.5) |
| Smoking status | | |
| Non-smoker | 29655 (47.8) | 3142 (48.8) |
| Ex-smoker | 17756 (28.6) | 1882 (29.2) |
| Light smoker (1–9 cigarettes/day) | 5931 (9.6) | 596 (9.3) |
| Moderate smoker (10–19 cigarettes/day) | 1125 (1.8) | 109 (1.7) |
| Heavy smoker (≥20 cigarettes/day) | 816 (1.3) | 76 (1.2) |
| Smoking not recorded before index date | 6720 (10.8) | 632 (9.8) |
| Alcohol consumption | | |
| Non-drinker | 31677 (51.1) | 3454 (53.7) |
| Trivial (<1 u/day) | 9215 (14.9) | 906 (14.1) |

**Table 2** Continued

| Characteristic | Study participants, number (column %) | |
| --- | --- | --- |
| | Controls (n=62 003) | Cases (n=6437) |
| Light (1–2 u/day) | 4460 (7.2) | 465 (7.2) |
| Moderate (3–6 u/day) | 4582 (7.4) | 453 (7) |
| Heavy (7–9 u/day) | 343 (0.6) | 30 (0.5) |
| Very Heavy (>9 u/day) | 129 (0.2) | 15 (0.2) |
| Alcohol not recorded before index date | 11 597 (18.7) | 1114 (17.3) |

Figures are numbers (%) unless otherwise indicated.
BMI, body mass index.

by Baxter and McDermott,[18] which explored the decision-making and referral process of GPs through semistructured interviews. The study found that among the 42 GPs who consented to participate, fasciculation was the most commonly described trigger for suspected MND and neurologist referral.[18] Our finding demonstrated that muscle fasciculation is indeed likely one of the top triggers for specialist referral practised by GPs across England. We found the patients who exhibited speech-related symptoms such as dysarthria received a diagnosis within an average of 6 months of symptom presentation. This suggests that the awareness of speech impairment in neurological disorders is relatively high among GPs in England, in contrast to previous studies.[18]

Signs and symptoms of respiratory and cognitive systems, while moderately associated with MND in univariate analysis, were no longer significant after accounting for symptoms related to bulbar region impairment and limb functions. The period between initial presentation of respiratory and cognitive symptoms and MND diagnosis is also significantly longer. This finding seems to indicate that presentations of respiratory or cognitive featured symptoms alone were insufficient to trigger a specialist referral. One reason for this may be that the symptom onset is relatively mild and sporadic and may not alarm the managing GPs since many other patients also have comorbidities with similar symptom profile. However, GPs could cross reference these symptoms against others (ie, dysarthria or muscle wasting) to enable neurologist referral.

The positive associations between constipation and urinary incontinence with MND found in our study were unexpected since these have not been described in MND before.[9 19] We suspect that the observed effect is primarily due to reverse causality: patients with undiagnosed but advanced MND are likely to have trouble with balance and movement, thereby hindering their ability to perform tasks such as going to the toilet.

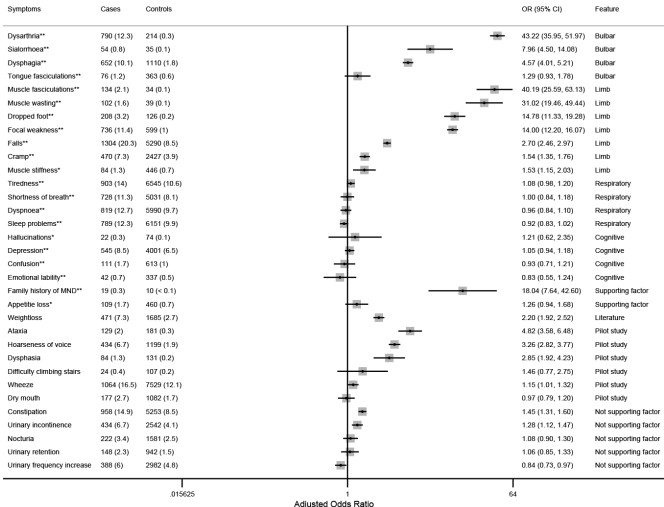

**Figure 1** Univariate analysis for risk of MND diagnosis based on 6437 cases and 62 003 age, sex, practice matched controls adjusted for variables shown, age, BMI, alcohol, smoking, ethnicity, deprivation. BMI, body mass index; MND, motor neuron disease.

**Figure 2** Multivariate analysis for risk of MND diagnosis based on 6437 cases and 62 003 age, sex, practice matched controls adjusted for variables shown, age, BMI, alcohol, smoking, ethnicity, deprivation. BMI, body mass index; MND, motor neuron disease.

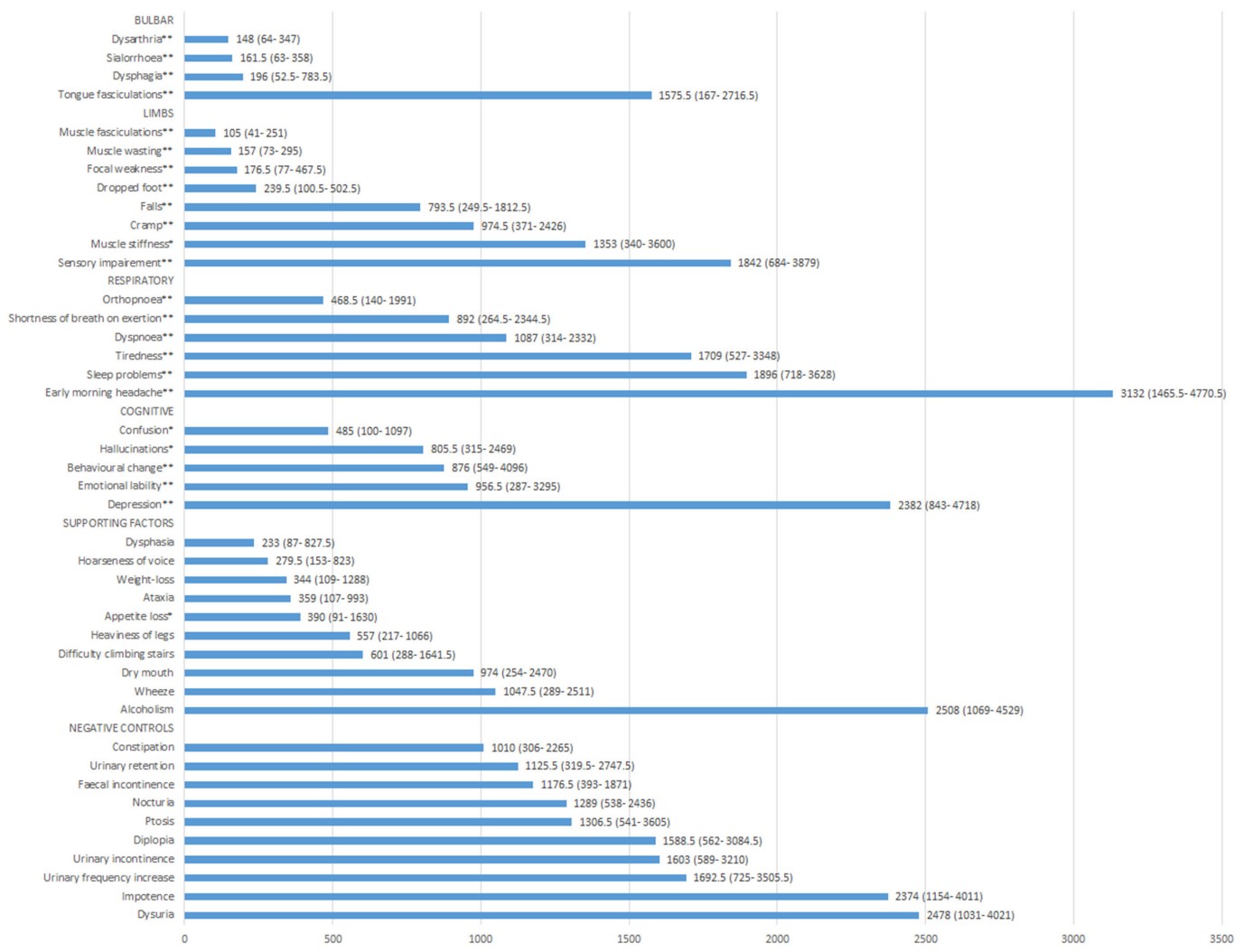

**Figure 3** Median duration (days, (IQR)) from earliest symptom presentation to MND diagnosis. MND, motor neuron disease.

### Strengths and limitations

This study is to date, the largest nested case–control study, derived from a representative population-based cohort looking at signs and symptoms related to MND. We were able to collate an extensive list of signs and symptoms from the GP records and to examine the association of each sign and symptom to subsequent MND diagnosis. We were also able to demonstrate that duration of the earliest symptom presentation to MND diagnosis could serve as an indicator to key signs and symptoms that triggered neurologist referral. We could not determine the time point, number and types of specialist referrals because our study was designed to compare the pattern of symptoms prior to diagnosis in cases (and the equivalent index date for controls). Another important limitation to our study is the completeness and accuracy of the recording of the signs and symptoms, since not all patients with symptoms will seek medical attention and not all symptoms will be recorded at the GP consultations. Furthermore, the positive associations may be attributed to unmeasured confounding or variation in coding practices at the GPs, due to the nature of routinely collected data. For example, we found dysphasia and wheeze to be independently and positively associated with MND. However, these symptoms are not traditionally considered within the clinical features of MND although dysphasia has been described to present in ALS in a systematic review.[20] It might also be that the patients presented with both dysarthria (disorder of speech) and dysphasia (disorder of language) but only dysphasia were coded because it was the dominant symptom when GP consultation was sought. Similarly, shortness of breath on exertion is a well-recognised sign of MND and might be coded as wheeze if symptom presentation of respiratory function tests is not undertaken. The association between ataxia and MND may be due to the fact that 'ataxia' is defined according to the clinical code used by GPs to describe what they thought the problem was when they examined the patient. As we now know that these patients subsequently were more likely to be diagnosed with MND, it is likely that what appeared to be 'ataxia' or 'imbalance due to muscle weakness' when the patient was examined by their GP, may actually have been due to MND.

## CONCLUSION

In this very large population-based nested case–control study of 6437 MND cases and 62 003 matched controls, we were able to confirm the currently recognised signs and symptoms, particularly those of bulbar and limb regions to be independently associated with MND. Additionally, we found GP-recorded hoarseness of voice, dysphasia, ataxia and weight loss to be associated with subsequent MND diagnosis. Due to the nature of the data as highlighted in the limitations, we could not verify whether all positive findings were true symptoms caused by MND or whether they were due to patient presentation patterns or GP coding behaviour. However, these findings have implications for GPs when considering referrals to neurologists and may be used to improve risk stratification and prediction of MND in primary care.

Future studies may focus on chronologically mapping the symptoms, characterising the patterns and clusters in symptoms presentation and predicting disease progression based on types and duration of existing symptoms.

**Contributors** XWM undertook data manipulation, reviewed the literature, led the data analysis, undertook interpretation of the data and wrote the first draft of the paper. JH-C led the study, obtained funding, data approvals, designed the study, drafted the protocol, contributed to the data management and interpretation of the data. CJM contributed to the funding application, contributed to protocol development and interpretation of results. CC contributed to the funding application, the discussion on protocol development and provided critical feedback on drafts of the manuscript. JB reviewed the literature on presentation of MND and conducted a search of how it had presented in patients in her own general practice. TAR contributed to interpretation of findings and compilation of the manuscript. XWM, JH-C, CC, CJM, JB, AR, and TAR approved the protocol, contributed to the critical revision of the manuscript and approved the final version of the manuscript. The corresponding author attests that all listed authors meet authorship criteria and that no others meeting the criteria have been omitted. JH-C accepts full responsibility for the work and/or the conduct of the study, had access to the data, and controlled the decision to publish.

**Funding** This study was funded by a grant from Motor Neurone Disease Association. The funders of this study contributed to the design and conduct of the study and reviewed and approve the manuscript. The funders had no role in analysis or interpretation of data. XM, TR, CC, JHC had full access to all the study data and JHC had final responsibility for submission.

**Competing interests** All authors have completed the ICMJE uniform disclosure form at www.icmje.org/coi_disclosure.pdf. JHC reports grants from grants from the John Fell Oxford University Press Research Fund; Cancer Research UK (CR-UK) grant number C5255/A18085, through the Cancer Research UK Oxford Centre and the Oxford Wellcome Institutional Strategic Support Fund (204826/Z/16/Z), during the conduct of the study. JHC is an unpaid director of QResearch, a not-for-profit organisation which is a partnership between the University of Oxford and EMIS Health who supply the QResearch database used for this work. JHC is a founder and shareholder of ClinRisk ltd and was its medical director until 31st May 2019. JHC is member of the SAGE subgroups on ethnicity and data and is chair of the risk stratification subgroup of NERVTAG. The views expressed are those of the authors only. CJM is supported by the NIHR Sheffield Biomedical Centre.

**Patient and public involvement** Patients and/or the public were involved in the design, or conduct, or reporting, or dissemination plans of this research. Refer to the Methods section for further details.

**Patient consent for publication** Not applicable.

**Ethics approval** The QResearch® ethics approval is with East Midlands-Derby Research Ethics Committee (reference 18/EM/0400, project reference OX1).

**Provenance and peer review** Not commissioned; externally peer reviewed.

**Data availability statement** Data may be obtained from a third party and are not publicly available. To guarantee the confidentiality of personal and health information only the authors have had access to the data during the study in accordance with the relevant licence agreements. Access to the QResearch data is according to the information on the QResearch website (www.qresearch.org).

**ORCID iDs**
Xue W Mei http://orcid.org/0000-0002-6279-4884
Tom A Ranger http://orcid.org/0000-0003-3091-2337
Christopher J McDermott http://orcid.org/0000-0002-1269-9053
Julia Hippisley-Cox http://orcid.org/0000-0002-2479-7283

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
