## [Reviewer comments · BMJ Open]

ARTICLE DETAILS

TITLE (PROVISIONAL)	Identifying Key Signs of Motor Neurone Disease in Primary Care: A Nested Case-Control Study Using the QResearch Database
AUTHORS	Mei, Xue; Burchardt, Judith; Ranger, Tom; McDermott, Christopher; Radunovic, Aleksandar; Coupland, Carol; Hippisley-Cox, Julia

VERSION 1 – REVIEW

REVIEWER	Logroscino, Giancarlo University of Bari, Department of Clinical Neurology and Research, Neurodegenerative Diseases Unit
REVIEW RETURNED	30-Jan-2022

GENERAL COMMENTS	Intro: A better division of the time spent before going to neurological care should be provided. This would be useful also to better understand the delay in primary care. A comparison of average time to specialist referral considering different set of symptoms should be reported. A comparison with data from population-based setting registries in Europe should be considered (Logroscino et al JNNP 2010). What is the average diagnostic delay in the study? Another additional feature would have been to report the information on the completeness of data collection of red flags symptoms and signs through the different practices involved in the study. Are there differences across practices based on geographic location of the practices in UK? Previous studies have shown that the onset (bulbar vs spinal) and other clinical features may vary across different geographic locations (Marin B. et al European Journal Epidemiology 2016). Do the authors have any information on the familiarity of ALS involved in the study?
---

REVIEWER	Izumi, Yuishin Tokushima University Hospital, Neurology
REVIEW RETURNED	26-Feb-2022

GENERAL COMMENTS	This is a clinically valuable paper. This study will help GP consult with a neurologist.
--

	Please clarify a following point. Instability due to muscle weakness is common in ALS patients. However cerebellar ataxia is generally not. The authors report that ataxia is an important finding of ALS. Please add an explanation of what the 'ataxia' means here.
--	---

REVIEWER	Faria, Christina Universidade Federal de Minas Gerais, Department of Physical Therapy
REVIEW RETURNED	14-Mar-2022

GENERAL COMMENTS	Dear authors, It was a pleasure to review this manuscript. I have identified its relevance as well as the importance of its results. I do believe that, before being suitable for publication, you must address the following points:  1) The aim described in the abstract is different from that described in the main text. Please, be consistent. 2) I do believe that this work was not planned to “to validate the symptoms and signs in the Red Flag tool”. For sure, important information was provided that can be used to the validation process of the “Red Flag tool”. Nowadays, we have recommendations to perform methodological studies which aimed to investigate the measurement properties of a tool: the COSMIN (COnsensus-based Standards for the selection of health Measurement INstruments). If you do believe that this study is a methodological study which aims to investigate the validity of the “Red Flag tool”, please, follow the COSMIN recommendation. If not, please, modify the points of the text related to the “validation”. 3) It is important to point out to if the final sample size was adequate to answer the study questions, mainly with so many variables analysed in the models. 4) In the Statistical analysis, please inform about the assumption of the statistical tests as well as about the significance level adopted in all statistical analysis. 5) It is mandatory to present the flow chart of the subjects in this type of study with so many subjects included and excluded. Readers must be able to follow this information in a clear and easy way. 6) It is important to provide the results of the statistical analyses considering the main parameters of the univariate and multivariate models.
--

VERSION 1 – AUTHOR RESPONSE

Reviewer: 1
Dr. Giancarlo Logroscino, University of Bari

Comments to the Author:

Intro: A better division of the time spent before going to neurological care should be provided. This would be useful also to better understand the delay in primary care.

Thank you. We have included a reference to the average diagnostic delay in MND diagnosis in the introduction.

A comparison of average time to specialist referral considering different set of symptoms should be reported. A comparison with data from population-based setting registries in Europe should be considered (Logroscino et al JNNP 2010). What is the average diagnostic delay in the study? We have included a comparison of incidence in the EU as suggested. We agree that it would be extremely helpful to recognise the drivers in primary care that lead to diagnostic delay. However, we cannot comment on delays since we could only capture time between first recorded symptoms and diagnosis. It is also difficult to know whether the symptom was directly related to the diagnosis/MND in many cases – for example, tiredness or depression may have had another cause. We have provided duration from first symptom presentation in the primary care to MND diagnosis (Figure 3), which may help us to understand the referral behaviours of GPs upon observation of certain particular symptoms. For example, patients with symptoms related to abnormal neurological functions such as dysarthria and muscle fasciculations received quickest diagnosis, indicating that those patients were referred to neurologists soon after symptom presentation at primary care.

Another additional feature would have been to report the information on the completeness of data collection of red flags symptoms and signs through the different practices involved in the study.

Are there differences across practices based on geographic location of the practices in UK? Previous studies have shown that the onset (bulbar vs spinal) and other clinical features may vary across different geographic locations (Marin B. et al European Journal Epidemiology 2016). We are not able to determine completeness of symptoms and signs with our data and have assumed that if a sign or symptom was not recorded then it was not present. However symptoms and signs in our study have been recorded electronically and prospectively via GP recording so we think it is reasonable to assume that the data is complete, although there may be some under-recording of signs and symptoms. We have already mentioned this as a limitation in the Discussion. There could be differences in clinical presentations of ALS/MND across geographical locations as your paper has pointed out. However, our study was designed to identify signs that could alert physicians in the primary care setting so we could not characterise clinical presentation of ALS/MND, the diagnosis of which needs to be made by neurologists.

Do the authors have any information on the familiarity of ALS involved in the study? By familiarity do you mean family history? If so we do have the information reported in Figures 1 and 2. There are in total 29 individuals with recorded family history of MND, with 19 (0.3%) of them being cases and 10 (<0.1%) being controls.

Reviewer: 2
Dr. Yuishin Izumi, Tokushima University Hospital

Comments to the Author:

This is a clinically valuable paper. This study will help GP consult with a neurologist.
Please clarify a following point.

Instability due to muscle weakness is common in ALS patients. However cerebellar ataxia is generally not. The authors report that ataxia is an important finding of ALS. Please add an explanation of what the 'ataxia' means here.

"Cerebellar ataxia" or "ataxia" is defined in this study according to the clinical code used by GPs to describe what they thought the problem was when they examined the patient. As we now know that these patients subsequently were more likely to be diagnosed with MND, it is likely that what appeared to be cerebellar ataxia when the patient was examined by their GP, may actually have been

due to MND. As our data on symptoms is from GP records, they are “investigatory” rather than “diagnostic”.

Reviewer: 3

Prof. Christina Faria, Universidade Federal de Minas Gerais

Comments to the Author:

Dear authors,

It was a pleasure to review this manuscript. I have identified its relevance as well as the importance of its results. I do believe that, before being suitable for publication, you must address the following points:

1) The aim described in the abstract is different from that described in the main text. Please, be consistent.

We have changed the aim in the abstract to be consistent with the main text.

2) I do believe that this work was not planned to “to validate the symptoms and signs in the Red Flag tool”. For sure, important information was provided that can be used to the validation process of the “Red Flag tool”. Nowadays, we have recommendations to perform methodological studies which aimed to investigate the measurement properties of a tool: the COSMIN (COnsensus-based Standards for the selection of health Measurement INstruments). If you do believe that this study is a methodological study which aims to investigate the validity of the “Red Flag tool”, please, follow the COSMIN recommendation. If not, please, modify the points of the text related to the “validation”.

Our study is not a methodological study and we did not aim to “validate” the Red Flag tool itself. As the Red Flag tool contains a collection of symptoms and signs recognised by neurologists, we aimed to verify and confirm the associations between those symptoms and signs and MND diagnosis. We have updated the term to “confirm” to clarify the context.

3) It is important to point out to if the final sample size was adequate to answer the study questions, mainly with so many variables analysed in the models.

We have added the sample size/power calculation to the Methods section of the paper.

4) In the Statistical analysis, please inform about the assumption of the statistical tests as well as about the significance level adopted in all statistical analysis.

We have added a summary of the assumptions and the significance level used in the last 2 sentences of the *statistical analysis* section in the Methods section. Our prespecified statistical significance was a two-tailed P-value of less than 0.05. We have also added the statistical software used in performing the analyses.

5) It is mandatory to present the flow chart of the subjects in this type of study with so many subjects included and excluded. Readers must be able to follow this information in a clear and easy way.

We identified all incident MND cases from a prospective cohort of individuals aged 18 years and over, during the period between 01/01/1998 and 31/12/2019. The matched controls were also identified from the same cohort. We have not applied any inclusion or exclusion criteria. However the specifics to case and control ascertainment can be found in *Ascertainment of cases and controls section in the METHOD section*.

6) It is important to provide the results of the statistical analyses considering the main parameters of the univariate and multivariate models.

We have presented both results in Figures 1 and 2.

VERSION 2 – REVIEW

REVIEWER	Logroscino, Giancarlo University of Bari, Department of Clinical Neurology and Research, Neurodegenerative Diseases Unit
REVIEW RETURNED	24-Apr-2022

GENERAL COMMENTS	This is an important study, describing the key role of GPs in the early diagnosis of MND. Minor comments: Some details on the importance of weight loss in the early diagnosis of MND should be added in the discussion. Ataxia and possible significance of this sign in the context of GB diagnosis of a neurological diseases should be analyzed.
--

VERSION 2 – AUTHOR RESPONSE

Reviewer Report:

Reviewer: 1

Dr. Giancarlo Logroscino, University of Bari

Comments to the Author:

This is an important study, describing the key role of GPs in the early diagnosis of MND.

Thank you

Minor comments:

Some details on the importance of weight loss in the early diagnosis of MND should be added in the discussion.

Thank you for this comment. While we appreciate that weight-loss is an important sign of possible MND, it is also a key sign to many other critical conditions such as many forms of cancers. We did not have information on weight-loss so we were not able to discuss further on that point.

Ataxia and possible significance of this sign in the context of GB diagnosis of a neurological diseases should be analyzed.

Thank you. GPs in the UK cannot formally diagnose neurological disorders like ataxia and MND. Since our diagnoses codes were obtained from GP records, we have added this as a limitation to our study.

Reviewer: 1

Competing interests of Reviewer: NA